# Underlying medical conditions and anti-SARS-CoV-2 spike IgG antibody titers after two doses of BNT162b2 vaccination: A cross-sectional study

Jiaqi Li [1]*, Takeshi Nakagawa [2], Masayo Kojima[1], Akihiko Nishikimi [3], Haruhiko Tokuda[4], Kunihiro Nishimura[5], Jun Umezawa [6], Shiori Tanaka[7], Manami Inoue [6,7], Norio Ohmagari[8], Koushi Yamaguchi[9], Kazuyoshi Takeda[10], Shohei Yamamoto [11], Maki Konishi[11], Kengo Miyo[12], Tetsuya Mizoue[11]

1 Department of Frailty Research, Research Institute, National Center for Geriatrics and Gerontology, Aichi, Japan, 2 Department of Social Science, Research Institute, National Center for Geriatrics and Gerontology, Aichi, Japan, 3 Biosafety Administration Division, Research Institute, National Center for Geriatrics and Gerontology, Aichi, Japan, 4 Department of Clinical Laboratory, Hospital, National Center for Geriatrics and Gerontology, Aichi, Japan, 5 Department of Preventive Medicine and Epidemiology, National Cerebral and Cardiovascular Center, Osaka, Japan, 6 Division of Cohort Research, Institute for Cancer Control, National Cancer Center, Tokyo, Japan, 7 Division of Prevention, Institute for Cancer Control, National Cancer Center, Tokyo, Japan, 8 Disease Control and Prevention Center, National Center for Global Health and Medicine, Tokyo, Japan, 9 Center for Maternal-Fetal, Neonatal and Reproductive Medicine, National Center for Child Health and Development, Tokyo, Japan, 10 Clinical Research & Education Promotion Division, National Center Hospital, National Center of Neurology and Psychiatry, Tokyo, Japan, 11 Department of Epidemiology and Prevention, Center for Clinical Sciences, National Center for Global Health and Medicine, Tokyo, Japan, 12 Center for Medical Informatics Intelligence, National Center for Global Health and Medicine, Tokyo, Japan

* hange.brucke@gmail.com

**Data Availability Statement:** The data is owned by the Department of Epidemiology and Prevention,

## Abstract

Patients with underlying medical conditions are at high risk of developing serious symptoms of the coronavirus disease 2019 than healthy individuals; therefore, it is necessary to evaluate the immune response to vaccination among them to formulate precision and personalized vaccination strategies. However, inconsistent evidence exists regarding whether patients with underlying medical conditions have lower anti-SARS-CoV-2 spike IgG antibody titers. We performed a cross-sectional study enrolling 2762 healthcare workers who received second doses of BNT162b2 vaccination from three medical and research institutes between June and July, 2021. Medical conditions were surveyed by a questionnaire, and spike IgG antibody titers were measured with chemiluminescent enzyme immunoassay using serum collected on the median of 62 days after the second vaccination. Multilevel linear regression model was used to estimate geometric mean and ratio of mean (95% confidence interval, CI) for the presence and absence of medical conditions and treatments. Among all participants (median age, 40 years [interquartile range, 30–50]; male proportion, 29.4%), the prevalence of hypertension, diabetes, chronic lung disease, cardiovascular disease, and cancer was 7.5%, 2.3%, 3.8%, 1.8%, and 1.3%, respectively. Patients with treated hypertension had lower antibody titers than those without hypertension; the multivariable-adjusted ratio of mean (95% CI) was 0.86 (0.76–0.98). Patients with untreated and treated diabetes had lower antibody titers than those without diabetes; the multivariable-adjusted ratio of mean (95% CI) was 0.63 (0.42–0.95) and 0.77 (0.63–

Center for Clinical Sciences, National Center for Global Health and Medicine, Japan, and is not publicly available due to ethical restrictions and participant confidentiality concerns, but de-identified data can be available by contacting the Department of Epidemiology and Prevention, Center for Clinical Sciences, National Center for Global Health and Medicine, Japan (website: http://epid.ncgm.go.jp/, email: yoboinfo@hosp.ncgm.go.jp, mizoue@hosp.ncgm.go.jp, tel: +81 3 3202 7181 [ext 2859]) for the researchers who meet the criteria for access to confidential data.

**Funding:** This study was supported by the Japan Health Research Promotion Bureau Research Fund (2020-B-09). The funder had no role in study design, data collection and analysis, decision to publish, or preparation of the manuscript.

**Competing interests:** The authors have declared that no competing interests exist.

0.95), respectively. No substantial difference was observed between the presence or absence of chronic lung disease, cardiovascular disease, or cancer. Patients with untreated hypertension and patients with untreated and treated diabetes had lower spike IgG antibody titers than participants without those medical conditions, suggesting that continuous monitoring of antibody titers and further booster shots could be necessary to maintain adaptive immunity in patients with hypertension or diabetes.

## Introduction

The coronavirus disease 2019 (COVID-19) pandemic is one of the most significant public health events in history [1]. Since the first official case was reported, the disease has affected over 644 million and caused the death of 6.6 million individuals worldwide up until December 2022 [2]. COVID-19 is highly contagious, but its mortality rate is not very high, as most cases of this disease are mild to moderate in nature. Severe symptoms were reported to be more common in patients with underlying medical conditions [3–5]. Accordingly, precautions for patients with underlying medical conditions received considerable attention. Messenger RNA based vaccine BNT162b2 is a vaccine used for active immunization to prevent COVID-19. It could elicit high SARS-CoV-2 neutralizing antibody titers and robust antigen-specific CD8+ and Th1-type CD4+ T-cell responses [6]. Clinical trials and observational studies have consistently demonstrated that messenger RNA (mRNA) based vaccines against COVID-19, such as BNT162b2, have an acceptable safety profile [7, 8], and two-dose vaccination of BNT162b2 has 95% (95% credible interval, 90.3–97.6) effective in preventing COVID-19 in persons aged 16 years or older [7]. Since 2021, BNT162b2 vaccine was used nationwide for COVID-19 prevention in Japan. Nevertheless, there have been several reports of poor immune response to vaccination in patients with underlying medical conditions. Several observational studies [9–11] reported that hypertension and diabetes were associated with lower spike IgG antibody titers following COVID-19 vaccination, while other studies did not [12–14]. Also, mixed findings were reported for other medical conditions such as chronic lung disease, cardiovascular disease, and cancer [10, 15–17].

Vaccination is a major measure employed to prevent the transmission of coronavirus and contain the COVID-19 pandemic [1, 3, 4]. It is important to evaluate the immune response to vaccination among high-risk groups such as patients with underlying medical conditions to establish precision and personalized vaccination strategies [18]. Given that previous inconsistent findings may be due to small sample size or single center study design, data from a multi-center large-scale population sample could provide critical evidence for evaluating the immune response to vaccination in patients with underlying medical conditions. Therefore, we aimed to investigate the association between underlying medical conditions and anti-SARS-CoV-2 spike IgG antibody titers in healthcare workers from national centers for advanced medical and research in Japan. We hypothesized that patients with underlying medical conditions have lower spike IgG antibody titers than those without these medical conditions after two doses of vaccination, and that antibody titers are different between patients with untreated and treated medical conditions.

## Materials and methods

### Study population

This cross-sectional study is part of a multicenter collaborative study targeting healthcare workers at six national centers for advanced medical and research in Japan. A questionnaire

was sent to healthcare workers to survey COVID-19-related information, and serum samples were collected during annual employee health checkups. Several participants from two national centers received three doses of vaccination, but they could not be identified because only the first or second doses of vaccination were recorded. One national center did not survey underlying medical conditions. Therefore, those national centers were excluded from this analysis. Totaling 4084 participants from three national centers finished the survey between June and July 2021. Of them, we excluded 445 who had less than two doses of vaccination, 208 who attended the survey within 14 days of the second vaccination, 2 who had extremely low spike IgG antibody titers (< 1 SU/mL), 19 who had infection history of COVID-19, 616 who did not report underlying medical conditions, and 32 who lacked data on covariates. After excluding those participants, 2762 participants aged 21 to 75 years were available. This study was performed in accordance with Declaration of Helsinki. Each participant provided written informed consent. The study design and procedure for data collection at each center were approved by the ethical committee of each center, while those of pooling study were approved by the National Center for Global Health and Medicine (approved number: NCGM-G-004233).

## Underlying medical conditions

Through a questionnaire, participants reported whether they currently had hypertension, diabetes, chronic lung disease, cardiovascular disease, or cancer, and whether they were receiving related treatments. Chronic lung disease was defined as chronic obstructive pulmonary disease, wheezing, or other persistent lung disorders. Cardiovascular disease was defined as ischemic heart disease or stroke.

## Covariates

Participants were asked about their occupation and lifestyle information, such as smoking, alcohol consumption, leisure-time physical activity, and sleep duration. Participants who smoked conventional cigarettes or used heated tobacco products daily were considered current smokers. Participants who consumed alcohol 1–2 days/week or more were considered weekly drinkers. Leisure-time physical activity, defined as physical activity including walking or gymnastics on holidays or when having free time, and sleeping duration were surveyed.

## Serological assay

We quantitatively measured SARS-CoV-2 IgG antibodies against spike protein in serum with the chemiluminescence enzyme immunoassay (CLEIA) platform (HISCL) manufactured by Sysmex Co. (Kobe, Japan) to assess vaccine-induced antibody response. HISCL was operated in a fully automatic manner using the chemiluminescent sandwich principle. A concentration of ≥10 SU/ml was considered seropositive [19]. More information on our serological assay has been reported [19]. This assay had a high correlation with the measured results using the AdviseDx SARS-CoV-2 IgG II assay (Abbott ARCHITECT®) in our prior analysis of 2961 participants, with a Spearman rank correlation coefficient of 0.96 (95% confidence interval, 0.95–0.96) [20].

Furthermore, we quantitatively tested SARS-CoV-2 IgG antibodies against nucleocapsid protein with the HISCL platform to assess past exposure of participants to SARS-CoV-2. The concentration of ≥10 SU/ml was considered seropositive, which has a sensitivity of 100% and specificity of 99.8% [19].

## Statistical analysis

Anti-SARS-CoV-2 spike IgG antibody titer distributions, including the geometric mean, were visualized using box plots based on the presence or absence of underlying medical conditions and treatments. Difference between the median IgG antibody titers were analyzed using the Kruskal-Wallis test. The Dunn's procedure adjusted by the Holm method was performed for multiple comparisons when the null hypothesis of the Kruskal-Wallis test was rejected. To control the influence of confounding factors and the different demographics of three national centers, we performed a multilevel linear regression analysis. Fixed effect covariates included age (years, continuous), sex (male or female), current smokers (no or yes), weekly drinkers (no or yes), occupation (doctor, nurse, allied healthcare professional, administrative staff, researcher, and others), body mass index ($kg/m^2$, continuous), an interaction term of sex and body mass index based on our previous findings [21], the interval between the second vaccination and blood sampling (days, continuous), the squared term of the interval, leisure-time physical activity (not engaged, 1–59, and $\geq$ 60 min/week), sleeping duration($<$ 6, 6–6.9, and $\geq$ 7 hours), and mutual adjustment for medical conditions. The random effects intercept term was set by national center levels. Considering spike IgG antibody titers obeyed a log-normal distribution, estimated marginal means for spike IgG antibody titers were calculated on a log-scale, and were back-transformed to present the geometric mean. The ratio of means (95% confidence intervals [CIs]) were calculated for multiple comparisons adjusted using the Dunnett's method. Statistical analyses were performed using R 4.2.1 (R Foundation for Statistical Computing, Vienna, Austria). A two-sided P value of $<$ 0.05 was considered statistically significant.

## Results

Participants' characteristics at three national centers are listed in Table 1. Overall, the median age was 40 years (interquartile range [IQR], 30–50), and the age proportion of 21 to 29, 30 to 39, 40 to 49, 50 to 59, 60 to 69, 70 to 75 years were 24.1%, 23.8%, 26.6%, 19.4%, 5.5%, and 0.5%. The male proportion was 29.4%. Most of the participants were medical staff, of whom 12.7% were doctors, 34.1% were nurses, 19.6% were allied health professionals, and the remaining third were mainly administrative staff and researchers. The prevalence of hypertension, diabetes, chronic lung disease, cardiovascular disease, and cancer were 7.5%, 2.3%, 3.8%, 1.8%, and 1.3%, respectively. The interval between the second dose of vaccine and antibody assay (IQR) was 62 days (36–69).

Spike IgG antibody titer distributions in non-patients, untreated patients, and treated patients are shown in Fig 1. The median spike IgG antibody titers in participants with treated hypertension (105 SU/mL; IQR, 59–191) was lower than that in participants without hypertension (169 SU/mL; IQR, 102–272), and this difference was statistically significant (P $<$ 0.001). Similarly, the median antibody titers in participants with treated diabetes (95 SU/mL; IQR, 60–171) was lower than that in participants without diabetes (167 SU/mL; IQR, 100–271), with a statistically significant difference (P $<$ 0.001). A borderline significant difference was observed in spike IgG antibody titers between participants with treated cardiovascular disease and those without cardiovascular disease (P = 0.08). However, in the post-hoc analysis, no substantial difference was noted in median antibody titers between participants with treated cardiovascular disease and those without cardiovascular disease (P = 0.34). Lastly, no substantial difference was noted in the median antibody titers between participants with or without chronic lung disease or cancer.

The estimated marginal means for geometric parameters of spike IgG antibody titers are listed in Table 2. After adjustments for age and sex, patients with treated hypertension had

**Table 1. Participants' characteristics according to three national centers.**

| | Total | National centers | | |
| --- | --- | --- | --- | --- |
| | | NCC | NCGG | NCGM |
| No. of participants | 2762 | 462 | 469 | 1831 |
| Male, % | 29.4 | 23.8 | 34.8 | 29.4 |
| Age, year | 40 (30–50) | 44 (35–52) | 41 (29–50) | 39 (29–49) |
| Occupation, % | | | | |
| doctor | 12.7 | 10.6 | 8.7 | 14.2 |
| nurse | 34.1 | 18.0 | 33.5 | 38.3 |
| allied health professional | 19.6 | 20.6 | 32.2 | 16.1 |
| administrative staff | 16.1 | 23.2 | 16.4 | 14.3 |
| researcher | 13.2 | 26.8 | 4.7 | 11.9 |
| other | 4.4 | 0.9 | 4.5 | 5.2 |
| Body mass index, kg/m$^2$ | 21.4 (19.6–23.6) | 21.5 (19.8–23.9) | 21.5 (19.9–23.4) | 21.2 (19.5–23.5) |
| Current smokers, % | 6.7 | 3.2 | 7.9 | 7.3 |
| Weekly drinkers*, % | 39.3 | 41.3 | 33.9 | 40.2 |
| Leisure-time physical activity, % | | | | |
| not engaged | 23.5 | 28.4 | 28.8 | 20.9 |
| 1–59 min/week | 41.6 | 29.4 | 42.4 | 44.5 |
| ≥ 60 min/week | 34.9 | 42.2 | 28.8 | 34.7 |
| Sleep duration, % | | | | |
| < 6 hours | 51.9 | 54.8 | 48.8 | 52.0 |
| 6–6.9 hours | 35.0 | 31.4 | 35.4 | 35.9 |
| ≥ 7 hours | 13.0 | 13.9 | 15.8 | 12.1 |
| Underlying medical conditions, % | | | | |
| hypertension | 7.5 | 9.1 | 8.1 | 7.0 |
| diabetes | 2.3 | 2.2 | 3.0 | 2.2 |
| chronic lung disease | 3.8 | 4.1 | 2.6 | 4.0 |
| cardiovascular disease | 1.8 | 1.9 | 2.3 | 1.7 |
| cancer | 1.3 | 2.4 | 0.6 | 1.2 |
| Interval between the second dose of vaccine and antibody assay, days | 62 (36–69) | 56 (32–63) | 33 (29–35) | 66 (56–70) |

Abbreviations: COPD, chronic obstructive pulmonary disease; NCC, National Cancer Center; NCGG, National Center for Geriatrics and Gerontology; NCGM, National Center for Global Medicine.

Values were presented as median (interquartile range) or proportion.

*Participants who drank 1–2 days/week or more were defined as weekly drinkers.

lower geometric mean spike IgG antibody titers than those without hypertension. Similar findings were observed in patients with treated diabetes. After further adjustments for other potential confounding factors, the multivariable-adjusted ratio of mean (95% CI) in patients with treated hypertension versus those without hypertension was 0.86 (0.76–0.98), and the multivariable-adjusted ratios of mean (95% CIs) in patients with untreated and treated diabetes versus those without diabetes were 0.63 (0.42–0.95) and 0.77 (0.63–0.95), respectively. No substantial difference in median spike IgG antibody titers was observed between the presence or absence of chronic lung disease, cardiovascular disease, or cancer.

## Discussion

In this multicenter collaborative study of 2762 participants who received two doses of vaccination, patients with untreated hypertension and patients with untreated and treated diabetes

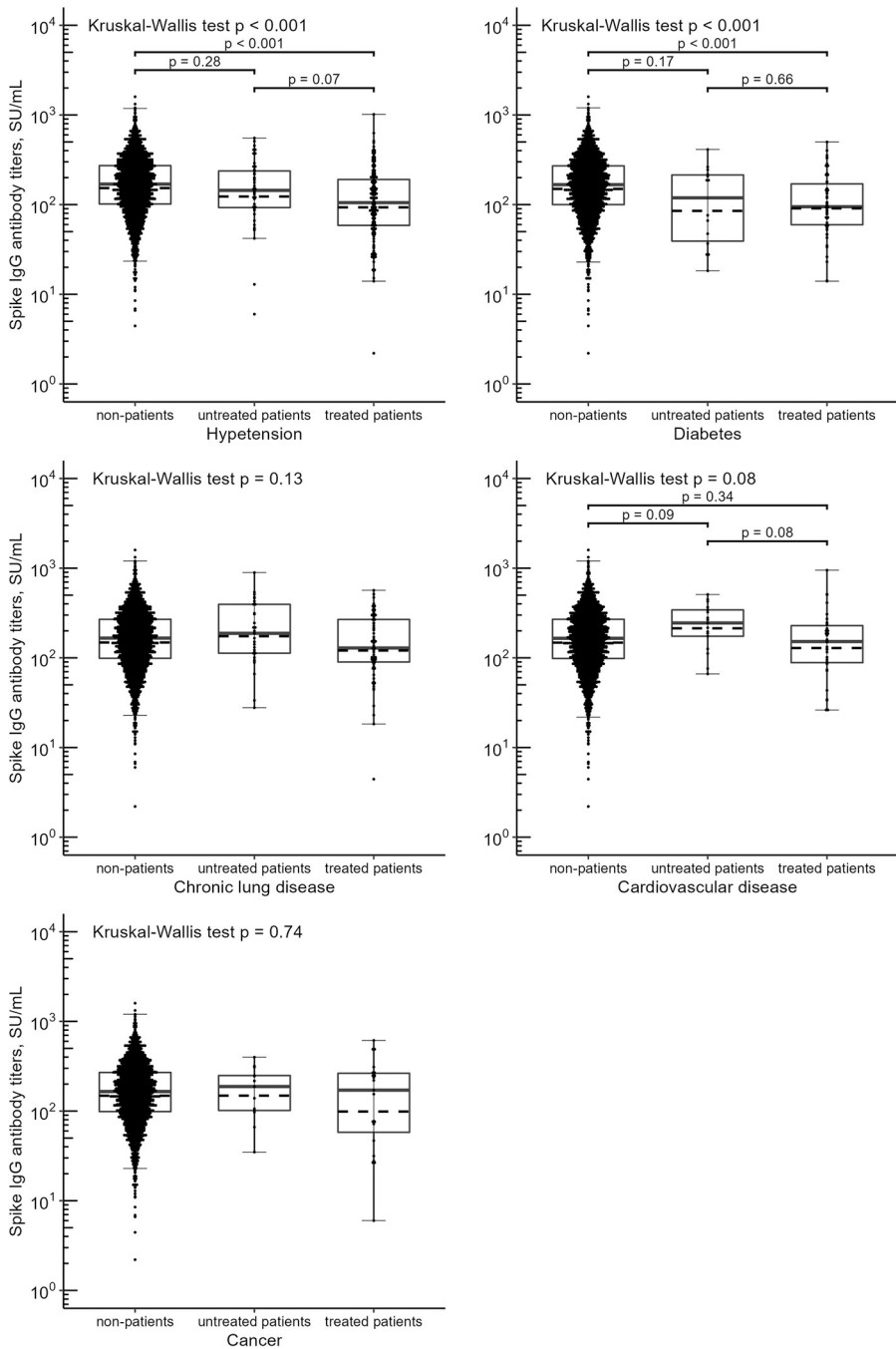

**Fig 1. Anti-SARS-CoV-2 spike IgG antibody titer distributions according to the presence or absence of chronic diseases and treatments.** The solid line in boxes represents the first quartile, the median value, and the third quartile. The dashed line represents the geometric mean value. The whiskers extend to the most extreme data point, which is no more than 1.5 times the length of interquartile range away from the box. Each black point represents an individual sample. Difference between the median of antibody titers was analyzed using the Kruskal-Wallis test. The Dunn's procedure adjusted by the Holm method was used for multiple comparisons when the null hypothesis of the Kruskal-Wallis test was rejected.

**Table 2. Estimated geometric means and 95% confidence intervals for anti-SARS-CoV-2 spike IgG antibody titers according to the presence or absence of underlying medical conditions and treatments.**

| | No. of participants | Sex- and age-adjusted model | | Multivariable-adjusted model | |
|---|---|---|---|---|---|
| | | Estimated geometric mean (95% CI) | Ratio of mean (95% CI) | Estimated geometric mean (95% CI) | Ratio of mean (95% CI) |
| Hypertension | | | | | |
| non-patients | 2554 | 155 (69–351) | 1.00 | 128 (87–189) | 1.00 |
| untreated patients | 44 | 156 (81–298) | 1.00 (0.79–1.27) | 118 (79–176) | 0.92 (0.74–1.15) |
| treated patients | 164 | 133 (64–279) | 0.86 (0.75–0.98) | 110 (75–163) | 0.86 (0.76–0.98) |
| Diabetes | | | | | |
| non-patients | 2698 | 155 (68–352) | 1.00 | 150 (102–222) | 1.00 |
| untreated patients | 14 | 116 (62–215) | 0.75 (0.50–1.13) | 95 (61–148) | 0.63 (0.42–0.95) |
| treated patients | 50 | 112 (58–218) | 0.73 (0.58–0.91) | 116 (78–174) | 0.77 (0.63–0.95) |
| Chronic lung disease | | | | | |
| non-patients | 2658 | 153 (68–346) | 1.00 | 121 (82–178) | 1.00 |
| untreated patients | 31 | 181 (97–338) | 1.18 (0.89–1.55) | 132 (88–197) | 1.09 (0.84–1.42) |
| treated patients | 73 | 145 (73–288) | 0.95 (0.79–1.14) | 105 (71–155) | 0.87 (0.74–1.02) |
| Cardiovascular disease | | | | | |
| non-patients | 2711 | 153 (68–345) | 1.00 | 102 (69–151) | 1.00 |
| untreated patients | 17 | 215 (117–396) | 1.41 (0.97–2.04) | 138 (90–212) | 1.35 (0.94–1.92) |
| treated patients | 34 | 185 (99–348) | 1.21 (0.93–1.59) | 118 (79–178) | 1.15 (0.90–1.47) |
| Cancer | | | | | |
| non-patients | 2726 | 153 (67–349) | 1.00 | 108 (73–160) | 1.00 |
| untreated patients | 13 | 207 (111–387) | 1.35 (0.88–2.08) | 143 (91–224) | 1.31 (0.87–1.99) |
| treated patients | 23 | 143 (77–265) | 0.93 (0.67–1.28) | 108 (71–165) | 1.00 (0.74–1.33) |

Abbreviations: CI, confidence interval.

The estimated marginal means for geometric parameters of spike IgG antibody titers were calculated using the multilevel linear regression model accounting for each national center as a random-effects intercept. The Dunnett's method was used for adjustment in multiple comparisons. The multivariable model was further adjusted for current smokers (no or yes), weekly drinkers (no or yes), occupation (doctor, nurse, allied healthcare professional, administrative staff, researcher, and other), body mass index (kg/m$^2$, continuous), an interaction term of sex and BMI, the interval between the second vaccination and blood sampling (days, continuous), the squared term of the interval, leisure-time physical activity (not engaged, 1–59, and $\geq 60$ min/week), sleeping duration($< 6$, 6–6.9, and $\geq 7$ hours), and mutual adjustment for medical conditions.

had lower spike IgG antibody titers than participants without those medical conditions. These findings provide evidence for the need to evaluate the immune response to vaccination in patients with hypertension or diabetes and are helpful for guiding future vaccination strategies.

A previous study reported the difference in spike IgG antibody titers measured from the first to fourth week after two doses of BNT162b2 vaccination in a sample of 86 healthcare workers; patients with hypertension had lower median antibody titers than those without hypertension (650 versus 1911 U/mL, P = 0.001) [9]. Similar difference was reported in a sample of 101 healthcare workers who measured antibody titers at the fifth month after two doses of inactivated vaccination (P = 0.01) [10]. However, two studies involving 248 and 712 healthcare workers reported similar antibody titers in participants with and without hypertension (P = 0.52 and 0.33, respectively; measured on the seventh day and third month after two doses of vaccination, respectively) [12, 13]. The mechanism underlying the weak immune response in patients with hypertension remains unclear. One hypothesis is that certain type of hypertension caused by the dysfunctional immune system [22, 23], in which case an inappropriate

response to vaccinations in patients with hypertension is not surprising. A previous study showed that patients with hypertension had lower lymphocytes counts and a higher neutrophil/lymphocyte ratio than those without hypertension, which supports above-mentioned hypothesis [10].

Persistently increased blood glucose and insulin resistance could suppress the immune system and increase the risk of getting infections [24, 25]. In a systematic review of eight studies, seven studies reported lower seropositivity of spike IgG antibody titers after two doses of vaccination in patients with diabetes than in those without diabetes. One study reported no association of diabetes with spike IgG antibody titers measured from the first to fourth week after two doses of vaccination (P = 0.88) [11]. Data from a recent study of 712 healthcare workers similarly reported no association for spike IgG antibody titers measured at the third month after two doses of vaccination (P = 0.77) [13]. Our multicenter collaborative study supports the inverse association between diabetes and spike IgG antibody titers and expand the association to patients with untreated and treated diabetes.

Our study observed similar spike IgG antibody titers in patients with chronic lung disease, cardiovascular disease, or cancer compared with patients without these conditions, which could be partly explained by the poor statistical power due to the small number of cases. A large-scale study of 1467 healthcare workers reported poor spike IgG antibody titers in patients with ischemic heart disease, but not in those with lung disease, compared with participants without those medical conditions at the first week after two doses of vaccination; the multivariable-adjusted ratios of mean (95% CIs) were 0.86 (0.75–0.99) and 1.09 (0.93–1.26), respectively [15]. In another case-control study of 593 patients with chronic pulmonary diseases and 593 controls, the prevalence of low responder, which was defined as neutralizing antibody index in the lowest quartile measured in the second month after two doses of vaccination, was higher in patients with chronic pulmonary diseases than in controls (34.7% versus 12.9%, p < 0.001) [16]. A meta-analysis of six studies showed that patients with solid tumors had adequate antibody response (> 90%), but the antibody titers were lower than those in participants without cancer [17].

The strength of this study is that we assessed the immune response based on the treatment status of underlying medical conditions in a large-scale survey. Several limitations of this study should be stated. First, medical conditions were surveyed using a self-reported questionnaire, which may not accurately identify the severity of medical conditions. Therefore, the difference in the immune response could be underestimated. However, healthcare workers had sufficient medical knowledge and underwent health checkups every year as a requirement of the Industrial Safety and Health Act; therefore, the possibility of false positives and false negatives should be small. Besides, participants reported other medical conditions such as rheumatoid arthritis, whereas we could not analyze them because of the limited case number. Second, the study population is healthcare workers, so that our findings may not representative for the general population. Third, the cross-sectional design is not optimal to investigate the time change in spike IgG antibody titers after vaccination in patients with underlying medical conditions. Fourth, we did not measure neutralizing antibodies, which is the gold standard assay for humoral immune response. However, spike IgG antibody titers measured with chemiluminescent enzyme immunoassay were reported to have good correlation with neutralizing antibody titers; the Spearman's rank correlation coefficient was 0.71 (95% CI, 0.63–0.77) [26]. Fifth, we did not assess the cellular immune response, which is another important mechanism of infection protection [27]. Sixth, the data of the third dose is still being compiled, so this study could not evaluate the antibody titers after the booster vaccination. Finally, we cannot rule out the possibility of residual confounding factors due to the observational study design.

## Conclusions

Patients with untreated hypertension and patients with untreated and treated diabetes had lower spike IgG antibody titers than participants without those medical conditions, suggesting that continuous monitoring of antibody titers and additional booster shots could be required to maintain adaptive immunity in patients with hypertension or diabetes.

## Acknowledgments

We thank Mika Shichishima for her contribution to data collection and administrative support.

## Author Contributions

**Conceptualization:** Jiaqi Li.

**Data curation:** Shohei Yamamoto, Maki Konishi.

**Formal analysis:** Jiaqi Li.

**Funding acquisition:** Tetsuya Mizoue.

**Investigation:** Takeshi Nakagawa, Masayo Kojima, Akihiko Nishikimi, Haruhiko Tokuda, Kunihiro Nishimura, Jun Umezawa, Shiori Tanaka, Manami Inoue, Norio Ohmagari, Koushi Yamaguchi, Kazuyoshi Takeda, Shohei Yamamoto, Maki Konishi, Tetsuya Mizoue.

**Methodology:** Jiaqi Li, Masayo Kojima, Kunihiro Nishimura, Manami Inoue, Koushi Yamaguchi, Tetsuya Mizoue.

**Project administration:** Tetsuya Mizoue.

**Resources:** Kengo Miyo.

**Supervision:** Takeshi Nakagawa, Masayo Kojima, Akihiko Nishikimi, Haruhiko Tokuda, Kunihiro Nishimura, Jun Umezawa, Shiori Tanaka, Manami Inoue, Norio Ohmagari, Koushi Yamaguchi, Kazuyoshi Takeda, Shohei Yamamoto, Maki Konishi, Kengo Miyo, Tetsuya Mizoue.

**Writing – original draft:** Jiaqi Li.

**Writing – review & editing:** Jiaqi Li, Takeshi Nakagawa, Masayo Kojima, Akihiko Nishikimi, Haruhiko Tokuda, Kunihiro Nishimura, Jun Umezawa, Shiori Tanaka, Manami Inoue, Norio Ohmagari, Koushi Yamaguchi, Kazuyoshi Takeda, Shohei Yamamoto, Maki Konishi, Kengo Miyo, Tetsuya Mizoue.

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
