## [Decision Letter · Decision Letter 0]

22 Nov 2022

PONE-D-22-29732Underlying medical conditions and anti-SARS-CoV-2 spike IgG antibody titers after two doses of BNT162b2 vaccination: a cross-sectional studyPLOS ONE

Dear Dr. Li,

Thank you for submitting your manuscript to PLOS ONE. After careful consideration, we feel that it has merit but does not fully meet PLOS ONE’s publication criteria as it currently stands. Therefore, we invite you to submit a revised version of the manuscript that addresses the points raised during the review process.

We look forward to receiving your revised manuscript.

Kind regards,

Hani Amir Aouissi, Ph.D.

Academic Editor

PLOS ONE

Journal Requirements:

"This study was supported by the Japan Health Research Promotion Bureau Research Fund (2020-B-09)."

"No"

Reviewers' comments:

Reviewer's Responses to Questions

**Comments to the Author**

1. Is the manuscript technically sound, and do the data support the conclusions?

Reviewer #1: Yes

2. Has the statistical analysis been performed appropriately and rigorously? 

Reviewer #1: No

3. Have the authors made all data underlying the findings in their manuscript fully available?

Reviewer #1: Yes

4. Is the manuscript presented in an intelligible fashion and written in standard English?

Reviewer #1: Yes

5. Review Comments to the Author

Reviewer #1: If you don’t mind I have some comments before acceptation, please correct as much as you can:

•Please start by a more general statement in the first sentence of the background in your abstract.

•I think you don’t have to specify (95% confidence interval, CI)

•Line 4-5 Please update data

•From line 1-7, there is a lack of references in the first part of your introduction, please cite more papers, you don’t have to limit yourself only to papers concerning your country, here are some papers to add:

1.https://doi.org/10.1111/cbdd.13761

2.https://doi.org/10.3390/healthcare10071341

3.https://doi.org/10.1186/s12941-021-00438-7

•There are also some very important studies concerning vaccination, its side effects, comparison between them and also booster doses, please add these two papers:

1.https://doi.org/10.3390/vaccines10111781

2. https://doi.org/10.3389/fpubh.2022.896343

•Please try to add at least 1-2 sentences in your introduction before your objective.

•I think it’s better to add a conclusion section instead of including it in the discussion section.

•Please put the first lines of your tables in bold

•My last comment is concerning English language, even if the paper is pleasant to read (specifically methods and discussion). However, there are some sentences that doesn’t read well. Please consider revising carefully your paper by a native English, a specialist or an English editing service.

I hope the authors will take my suggestions into account and revise their manuscript.

6. PLOS authors have the option to publish the peer review history of their article (what does this mean?). If published, this will include your full peer review and any attached files.

Reviewer #1: No

---

## [Author Response · Author response to Decision Letter 0]

19 Dec 2022

Manuscript ID: PONE-D-22-29732

Underlying medical conditions and anti-SARS-CoV-2 spike IgG antibody titers after two doses of BNT162b2 vaccination: a cross-sectional study

PLOS One

Many thanks for your thoughtful reviews and positive evaluation for our study. We have revised our manuscript in response to your suggestions, and used English editing services to improve the manuscript English quality. Our responses are described in normal font following the reviewers’ comments in boldface. All modifications in the current version of manuscript are highlighted in a font color of red. We hope that our revision will be satisfactory for your publication criteria.

Reviewer #1:

 If you don’t mind I have some comments before acceptation, please correct as much as you can:

1. Please start by a more general statement in the first sentence of the background in your abstract.

Response: 

Thank you very much for your insightful comments. We have added a general statement in the abstract.

Lines 1 to 5 in the Abstract.

“Patients with underlying medical conditions are at high risk of developing serious symptoms of the coronavirus disease 2019 than healthy individuals, so that it is necessary to evaluate the immune response to vaccination among them to formulate precision and personalized vaccination strategies. However, inconsistent evidence exists regarding whether patients with underlying medical conditions have lower anti-SARS-CoV-2 spike IgG antibody titers.”

2. I think you don’t have to specify (95% confidence interval, CI)

Response: 

We have discussed your suggestion, but we think that 95% confidence interval is necessary to show the interval of the true population mean at a 5% level (p < 0.05). We believe it could provide effective information for readers to understand the results. For reference, the following Covid antibodies-related manuscript showed the 95% confidence interval of ratio of mean. 

DOI: 10.1056/NEJMoa2114583

3. Line 4-5 Please update data

Response: 

We have updated the data on the epidemic status of COVID-19.

Line 3 to 5, page 5.

“Since the first official case was reported, the disease has affected over 644 million and caused the death of 6.6 million individuals worldwide up until December 2022 [2].”

4. From line 1-7, there is a lack of references in the first part of your introduction, please cite more papers, you don’t have to limit yourself only to papers concerning your country, here are some papers to add:

1.https://doi.org/10.1111/cbdd.13761

2.https://doi.org/10.3390/healthcare10071341

3.https://doi.org/10.1186/s12941-021-00438-7

Response: 

We have added more related studies as references in the introduction, as suggested.

5. There are also some very important studies concerning vaccination, its side effects, comparison between them and also booster doses, please add these two papers:

1. https://doi.org/10.3390/vaccines10111781

2. https://doi.org/10.3389/fpubh.2022.896343

Response: 

These papers are about inactivated-virus and adenoviral-vector vaccines, while participants received messenger RNA vaccine in our study. Also, we did not discuss mild side effects (pain, fatigue, fever, and etc.) of vaccination, our concern is the difference of IgG antibody titers in different population. So, we did not add these studies as additional references.

6. Please try to add at least 1-2 sentences in your introduction before your objective.

Response: 

We have modified the description of the purpose of this study for clarity in Introduction.

Line 17 to 26, page 5 to 6.

“Vaccination is a major measure employed to prevent the transmission of coronavirus and contain the COVID-19 pandemic. It is important to evaluate the immune response to vaccination among high-risk groups such as patients with underlying medical conditions to establish precision and personalized vaccination strategies. Given that previous inconsistent findings may be due to small sample size or single center study design, data from a multicenter large-scale population sample could provide critical evidence for evaluating the immune response to vaccination in patients with underlying medical conditions. Therefore, we aimed to investigate the association between underlying medical conditions and anti-SARS-CoV-2 spike IgG antibody titers in healthcare workers from national centers for advanced medical and research in Japan.”

7. I think it’s better to add a conclusion section instead of including it in the discussion section.

Response: 

We have added a conclusion section as you suggested.

8. Please put the first lines of your tables in bold

Response: 

We have put in bold the title of all tables.

9. My last comment is concerning English language, even if the paper is pleasant to read (specifically methods and discussion). However, there are some sentences that doesn’t read well. Please consider revising carefully your paper by a native English, a specialist or an English editing service.

Response: 

We have used an English editing service to improve the English quality in the current version of our manuscript.

---

## [Decision Letter · Decision Letter 1]

5 Feb 2023

PONE-D-22-29732R1Underlying medical conditions and anti-SARS-CoV-2 spike IgG antibody titers after two doses of BNT162b2 vaccination: a cross-sectional studyPLOS ONE

Dear Dr. Li,

Thank you for submitting your manuscript to PLOS ONE. After careful consideration, we feel that it has merit but does not fully meet PLOS ONE’s publication criteria as it currently stands. Therefore, we invite you to submit a revised version of the manuscript that addresses the points raised during the review process.

We look forward to receiving your revised manuscript.

Kind regards,

Hani Amir Aouissi, Ph.D.

Academic Editor

PLOS ONE

Reviewers' comments:

Reviewer's Responses to Questions

**Comments to the Author**

1. If the authors have adequately addressed your comments raised in a previous round of review and you feel that this manuscript is now acceptable for publication, you may indicate that here to bypass the “Comments to the Author” section, enter your conflict of interest statement in the “Confidential to Editor” section, and submit your "Accept" recommendation.

Reviewer #1: All comments have been addressed

Reviewer #2: All comments have been addressed

2. Is the manuscript technically sound, and do the data support the conclusions?

Reviewer #1: Yes

Reviewer #2: Partly

3. Has the statistical analysis been performed appropriately and rigorously? 

Reviewer #1: Yes

Reviewer #2: Yes

4. Have the authors made all data underlying the findings in their manuscript fully available?

Reviewer #1: Yes

Reviewer #2: No

5. Is the manuscript presented in an intelligible fashion and written in standard English?

Reviewer #1: Yes

Reviewer #2: Yes

6. Review Comments to the Author

Reviewer #1: We thank you for your great work.

Have you been working on other diseases such as arthritis? eyes illnesses...

Another point, if you find a person who has all the diseases that you worked on, what is the diagnosis of your study in this case.

Why didn't you work on samples from the general public?

Outline a short introduction to BNT162b2

I think the first infection with BNT162b2 was in 2021!!!

What are the methods of prevention and protection that were used in the study? For you and also for the treated patients.

Where I found your work very important and must work on developing it in the future.

Reviewer #2: This study compared spike IgG antibody titers in health care workers with and without disease among subjects who received second doses of BNT162b2 vaccination. The major issues are as follows:

1. Antibody titers have been shown to change with booster doses. And since booster doses are the standard in the US and elsewhere, antibody titers will likely change. Antibody titers after booster vaccination should also be presented in the results.

2. There is not correlate of protection defined for any COVID-19 vaccine so although levels decay, we don't know what level would leave you vulnerable to infection. Efficacy measures would be more important than antibody titers. This has rapidly changed with emerging variants. Vaccine protection to the original strain is not the same as Omicron etc.

3. Since this study was conducted in healthcare workers, who have very high opportunities for exposure to COVID-19, the presence or absence of COVID-19 infection should be clearly identified. In addition, the presence or absence of COVID-19 infection should be included as a covariate in the adjusted analysis.

4. The antibody titers were measured to health care workers in hospitals, which is not representative for the general population (at least not all age groups I suppose). It would be better if the authors report the age distribution of the participants, which would help with interpreting the results.

5. The authors may also need to provide more details about the assay methods for measuring the antibodies, such as the type of antibodies (spike or RBD binding antibody or neutralising antibody) and the tested strain.

7. PLOS authors have the option to publish the peer review history of their article (what does this mean?). If published, this will include your full peer review and any attached files.

Reviewer #1: No

Reviewer #2: No

---

## [Author Response · Author response to Decision Letter 1]

16 Feb 2023

Manuscript ID: PONE-D-22-29732R1

Underlying medical conditions and anti-SARS-CoV-2 spike IgG antibody titers after two doses of BNT162b2 vaccination: a cross-sectional study

PLOS ONE

Many thanks for your thoughtful reviews and positive evaluation for our study. We have revised our manuscript in response to reviewers’ suggestions. Our responses are described in normal font following the reviewers’ comments in boldface. All modifications in the current version of manuscript are highlighted in a font color of red. We hope that our revision will be satisfactory for your publication criteria.

Reviewer #1: We thank you for your great work.

1. Have you been working on other diseases such as arthritis? eyes illnesses...

Response: 

Thank you very much for your favorable comments. A small number of study participants reported other conditions including rheumatoid arthritis and glaucoma (no other eye diseases). The number of these other conditions is very small (5 for rheumatoid arthritis and 6 for glaucoma), so we did not analyze or adjust other conditions. We have added it as a limitation.

Line 187-189

“Besides, participants reported other medical conditions such as rheumatoid arthritis, whereas we could not analyze them because of the limited case number.”

2. Another point, if you find a person who has all the diseases that you worked on, what is the diagnosis of your study in this case.

Response: 

We have controlled the impact of that situation by performing mutual adjustment for these medical conditions (Line 94-95).

3. Why didn't you work on samples from the general public?

Response: 

Healthcare workers have a higher proportion of vaccinations, and the participation rate of questionnaire surveys is relatively high, and blood samples are easier to collect compared to the general population. We have added this as a limitation.

Line 189-190

“Second, the study population is healthcare workers, so that our findings may not representative for the general population.”

4. Outline a short introduction to BNT162b2.

Response: 

We have added more introduction on BNT162b2 in the Introduction as following. 

Line 8 to 14:

Messenger RNA based vaccine BNT162b2 is a vaccine used for active immunization to prevent COVID-19. It could elicit high SARS-CoV-2 neutralizing antibody titers and robust antigen-specific CD8+ and Th1-type CD4+ T-cell responses [6]. Clinical trials and observational studies have consistently demonstrated that BNT162b2 have an acceptable safety profile [7, 8], and two-dose vaccination of BNT162b2 has 95% (95% credible interval, 90.3-97.6) effective in preventing COVID-19 in persons aged 16 years or older [6].

5. I think the first infection with BNT162b2 was in 2021!!!

Response: 

We have added the following sentence in the Introduction.

Line 14-15:

“Since 2021, BNT162b2 vaccine was used nationwide for COVID-19 prevention in Japan.”

6. What are the methods of prevention and protection that were used in the study? For you and also for the treated patients.

Response: 

Preventive measures for COVID include wearing a mask, washing hands and sanitizing with alcohol frequently, practicing proper cough and sneeze etiquette, avoiding the "closed spaces with poor ventilation," "crowded spaces with many people nearby," and "close-contact settings such as close-range conversations" and more. Those preventive measures do not directly mediate the association between diseases and antibody titers, so we did not add them in the revised manuscript.

7. Where I found your work very important and must work on developing it in the future.

Response: 

The study project is continuing to evaluate vaccine boosters and their long-term impacts. We are committed to providing more valuable data for infectious disease prevention policies.

Reviewer #2: This study compared spike IgG antibody titers in health care workers with and without disease among subjects who received second doses of BNT162b2 vaccination. The major issues are as follows:

1. Antibody titers have been shown to change with booster doses. And since booster doses are the standard in the US and elsewhere, antibody titers will likely change. Antibody titers after booster vaccination should also be presented in the results.

Response: 

Thank you very much for your favorable comments. This data is a report for assessing antibody titers after the second vaccination. The data set of vaccine boosters is still being sorted out. We will report the data on booster doses in an upcoming study. We have added this as a limitation.

Line 197-198

“Sixth, the data of the third dose is still being compiled, so this study could not evaluate the antibody titers after the booster vaccination.”

2. There is not correlate of protection defined for any COVID-19 vaccine so although levels decay, we don't know what level would leave you vulnerable to infection. Efficacy measures would be more important than antibody titers. This has rapidly changed with emerging variants. Vaccine protection to the original strain is not the same as Omicron etc.

Response: 

As you said, vaccine efficacy is important and may change as new variant emerges. Our study is an observational study and the study population had a very low incidence of Covid-19. Therefore, it is difficult for our study to provide data on vaccine efficacy assessment. The aim of the current study is to evaluate antibody titers after two doses vaccination.

A concentration of ≥10 SU/ml was considered seropositive for the measured antibody titers in our study . In an ongoing multinational, placebo-controlled, observer-blinded, pivotal efficacy trial, BNT162b2 was 95% effective in preventing Covid-19 (95% credible interval, 90.3 to 97.6). We have added the information in the revised manuscript.

Line 67 to 78:

“We quantitatively measured SARS-CoV-2 IgG antibodies against spike protein in serum with the chemiluminescence enzyme immunoassay (CLEIA) platform (HISCL) manufactured by Sysmex Co. (Kobe, Japan) to assess vaccine-induced antibody response. HISCL was operated in a fully automatic manner using the chemiluminescent sandwich principle. A concentration of ≥10 SU/ml was considered seropositive [19]. More information on our serological assay has been reported [19]. This assay had a high correlation with the measured results using the AdviseDx SARS-CoV-2 IgG II assay (Abbott ARCHITECT®) in our prior analysis of 2961 participants, with a Spearman rank correlation coefficient of 0.96 (95% confidence interval, 0.95–0.96) [20].

Furthermore, we quantitatively tested SARS-CoV-2 IgG antibodies against nucleocapsid protein with the HISCL platform to assess past exposure of participants to SARS-CoV-2. The concentration of ≥10 SU/ml was considered seropositive, which has a sensitivity of 100% and specificity of 99.8% [19]. ”

Line 11 to 14:

“Clinical trials and observational studies have consistently demonstrated that BNT162b2 have an acceptable safety profile [7, 8], and two-dose vaccination of BNT162b2 has 95% (95% credible interval, 90.3-97.6) effective in preventing COVID-19 in persons aged 16 years or older [7].”

3. Since this study was conducted in healthcare workers, who have very high opportunities for exposure to COVID-19, the presence or absence of COVID-19 infection should be clearly identified. In addition, the presence or absence of COVID-19 infection should be included as a covariate in the adjusted analysis.

Response: 

The number of participants with an infection history of COVID-19 is very small (19 participants) in this study. Consistently, our previous study reported that the seroprevalence in healthcare workers was 0.67% (95% CI, 0.29%-1.32%) between August and October, 2020 [a]. We have excluded these participants with an infection history of COVID-19 as we described in the Methods (Line 45-46). 

a. Yazaki S, Yoshida T, Kojima Y, et al. Difference in SARS-CoV-2 Antibody Status Between Patients With Cancer and Health Care Workers During the COVID-19 Pandemic in Japan. JAMA Oncol. 2021;7(8):1141-1148. 

4. The antibody titers were measured to health care workers in hospitals, which is not representative for the general population (at least not all age groups I suppose). It would be better if the authors report the age distribution of the participants, which would help with interpreting the results.

Response: 

The study participants aged 21 to 75 years as we described in the Methods (Line 43). Moreover, we have added age proportion in the Results.

Line 104-105:

“Overall, the median age was 40 years (interquartile range [IQR], 30–50), and the age proportion of 21 to 29, 30 to 39, 40 to 49, 50 to 59, 60 to 69, 70 to 75 years were 24.1%, 23.8%, 26.6%, 19.4%, 5.5%, and 0.5%.”

Also, we have added this as a limitation.

Line 189-190

“Second, the study population is healthcare workers, so that our findings may not representative for the general population.”

5. The authors may also need to provide more details about the assay methods for measuring the antibodies, such as the type of antibodies (spike or RBD binding antibody or neutralising antibody) and the tested strain.

Response: 

The type of antibody we measured is spike IgG antibody. We have added more information in the revised manuscript. 

Line 67 to 78:

“We quantitatively measured SARS-CoV-2 IgG antibodies against spike protein in serum with the chemiluminescence enzyme immunoassay (CLEIA) platform (HISCL) manufactured by Sysmex Co. (Kobe, Japan) to assess vaccine-induced antibody response. HISCL was operated in a fully automatic manner using the chemiluminescent sandwich principle. A concentration of ≥10 SU/ml was considered seropositive [19]. More information on our serological assay has been reported [19]. This assay had a high correlation with the measured results using the AdviseDx SARS-CoV-2 IgG II assay (Abbott ARCHITECT®) in our prior analysis of 2961 participants, with a Spearman rank correlation coefficient of 0.96 (95% confidence interval, 0.95–0.96) [20].

Furthermore, we quantitatively tested SARS-CoV-2 IgG antibodies against nucleocapsid protein with the HISCL platform to assess past exposure of participants to SARS-CoV-2. The concentration of ≥10 SU/ml was considered seropositive, which has a sensitivity of 100% and specificity of 99.8% [19]. ”

---

## [Decision Letter · Decision Letter 2]

14 Mar 2023

Underlying medical conditions and anti-SARS-CoV-2 spike IgG antibody titers after two doses of BNT162b2 vaccination: a cross-sectional study

PONE-D-22-29732R2

Dear Dr. Li,

We’re pleased to inform you that your manuscript has been judged scientifically suitable for publication and will be formally accepted for publication once it meets all outstanding technical requirements.

Kind regards,

Hani Amir Aouissi, Ph.D.

Academic Editor

PLOS ONE

Additional Editor Comments (optional):

Dear authors, thank you for the great effort you put in this revised version.

According to the reviewers you adressed almost all the comments. From my side I have no more comment.

I think the paper is suitable for further processing.

Reviewers' comments:

Reviewer's Responses to Questions

**Comments to the Author**

1. If the authors have adequately addressed your comments raised in a previous round of review and you feel that this manuscript is now acceptable for publication, you may indicate that here to bypass the “Comments to the Author” section, enter your conflict of interest statement in the “Confidential to Editor” section, and submit your "Accept" recommendation.

Reviewer #1: All comments have been addressed

Reviewer #2: All comments have been addressed

2. Is the manuscript technically sound, and do the data support the conclusions?

Reviewer #1: Yes

Reviewer #2: Yes

3. Has the statistical analysis been performed appropriately and rigorously? 

Reviewer #1: Yes

Reviewer #2: Yes

4. Have the authors made all data underlying the findings in their manuscript fully available?

Reviewer #1: Yes

Reviewer #2: Yes

5. Is the manuscript presented in an intelligible fashion and written in standard English?

Reviewer #1: Yes

Reviewer #2: Yes

6. Review Comments to the Author

Reviewer #1: The work of the presenter is respected. It deserves to be published. and We hope to see you in other works.

Reviewer #2: (No Response)

7. PLOS authors have the option to publish the peer review history of their article (what does this mean?). If published, this will include your full peer review and any attached files.

Reviewer #1: No

Reviewer #2: No

---

## [Editor Report · Acceptance letter]

28 Mar 2023

PONE-D-22-29732R2 

Underlying medical conditions and anti-SARS-CoV-2 spike IgG antibody titers after two doses of BNT162b2 vaccination: a cross-sectional study 

Dear Dr. Li:

I'm pleased to inform you that your manuscript has been deemed suitable for publication in PLOS ONE. Congratulations! Your manuscript is now with our production department. 

Kind regards, 

on behalf of

Dr. Hani Amir Aouissi 

Academic Editor

PLOS ONE